# The role of hospital managers in quality and patient safety: a systematic review

Anam Parand,[1] Sue Dopson,[2] Anna Renz,[1] Charles Vincent[3]

▶ Prepublication history and additional material for is available. To view please visit the journal (http://dx.doi.org/10.1136/bmjopen-2014-005055).

[1]Department of Surgery & Cancer, Imperial College London, London, UK
[2]Said Business School, University of Oxford, Oxford, UK
[3]Department of Experimental Psychology, University of Oxford, Oxford, UK

**Correspondence to**
Dr Anam Parand;
a.parand@imperial.ac.uk

## ABSTRACT

**Objectives:** To review the empirical literature to identify the activities, time spent and engagement of hospital managers in quality of care.

**Design:** A systematic review of the literature.

**Methods:** A search was carried out on the databases MEDLINE, PSYCHINFO, EMBASE, HMIC. The search strategy covered three facets: management, quality of care and the hospital setting comprising medical subject headings and key terms. Reviewers screened 15 447 titles/abstracts and 423 full texts were checked against inclusion criteria. Data extraction and quality assessment were performed on 19 included articles.

**Results:** The majority of studies were set in the USA and investigated Board/senior level management. The most common research designs were interviews and surveys on the perceptions of managerial quality and safety practices. Managerial activities comprised strategy, culture and data-centred activities, such as driving improvement culture and promotion of quality, strategy/goal setting and providing feedback. Significant positive associations with quality included compensation attached to quality, using quality improvement measures and having a Board quality committee. However, there is an inconsistency and inadequate employment of these conditions and actions across the sample hospitals.

**Conclusions:** There is some evidence that managers' time spent and work can influence quality and safety clinical outcomes, processes and performance. However, there is a dearth of empirical studies, further weakened by a lack of objective outcome measures and little examination of actual actions undertaken. We present a model to summarise the conditions and activities that affect quality performance.

## INTRODUCTION

Managers in healthcare have a legal and moral obligation to ensure a high quality of patient care and to strive to improve care. These managers are in a prime position to mandate policy, systems, procedures and organisational climates. Accordingly, many have argued that it is evident that healthcare managers possess an important and obvious role in quality of care and patient safety and that it is one of the highest priorities of

### Strengths and limitations of this study

- This is the first systematic review of the literature that has considered the evidence on Boards' and managers' time spent, engagement and work within the context of quality and safety. This review adds to the widely anecdotal and commentary pieces that speculate on what managers should be doing by presenting what they are actually doing.
- The review reveals conditions and actions conducive to good quality management and offers a model to transparently present these to managers considering their own part in quality and safety.
- The search for this review has screened a vast amount of the literature (over 15 000 articles) across a number of databases.
- The small number of included studies and their varied study aims, design and population samples make generalisations difficult. With more literature on this topic, distinctions could be made between job positions.
- The quality assessment scores are subjective and may not take into consideration factors beyond the quality assessment scale used.

healthcare managers.[1–3] In line with this, there have been calls for Boards to take responsibility for quality and safety outcomes.[4 5] One article warned hospital leaders of the dangers of following in the path of bankers falling into recession, constrained by their lack of risk awareness and reluctance to take responsibility.[6] To add to the momentum are some high profile publicity of hospital management failures affecting quality and safety, eliciting strong instruction for managerial leadership for quality at the national level in some countries.[7 8]

Beyond healthcare, there is clear evidence of managerial impact on workplace safety.[9–12] Within the literature on healthcare, there are non-empirical articles providing propositions and descriptions on managerial attitudes and efforts to improve safety and quality. This literature, made up of opinion articles, editorials and single participant experiences, present an array of insightful suggestions and recommendations for actions that hospital

managers should take to improve the quality of patient care delivery in their organisation.[13–17] However, researchers have indicated that there is a limited evidence base on this topic.[18–21] Others highlight the literature focus on the difficulties of the managers' role and the negative results of poor leadership on quality improvement (QI) rather than considering actions that managers presently undertake on quality and safety.[22 23] Consequently, little is known about what healthcare managers are doing in practice to ensure and improve quality of care and patient safety, how much time they spend on this, and what research-based guidance is available for managers in order for them to decide on appropriate areas to become involved. Due perhaps to the broad nature of the topic, scientific studies exploring these acts and their impact are likely to be a methodological challenge, although a systematic review of the evidence on this subject is notably absent. This present systematic literature review aims to identify empirical studies pertaining to the role of hospital managers in quality of care and patient safety. We define 'role' to comprise of managerial activities, time spent and active engagement in quality and safety and its improvement. While the primary research question is on the managers' role, we take into consideration the contextual factors surrounding this role and its impact or importance as highlighted by the included studies. Our overarching question is "What is the role of hospital managers in quality and safety and its improvement?" The specific review research questions are as follows:

▶ How much time is spent by hospital managers on quality and safety and its improvement?
▶ What are the managerial activities that relate to quality and safety and its improvement?
▶ How are managers engaged in quality and safety and its improvement?
▶ What impact do managers have on quality and safety and its improvement?
▶ How do contextual factors influence the managers' role and impact on quality and safety and its improvement?

## METHODS
### Concepts and definitions
Quality of care and patient safety were defined on the basis of widely accepted definitions from the Institute of Medicine (IOM) and the Agency for Healthcare Research and Quality Patient Safety Network (AHRQ PSN). IOM define quality in healthcare as possessing the following dimensions: safe, effective, patient-centred, timely, efficient and equitable.[4] They define patient safety simply as "the prevention of harm to patients",[24] and AHRQ define it as "freedom from accidental or preventable injuries produced by medical care."[25] Literature was searched for all key terms associated with quality and patient safety to produce an all-encompassing approach. A manager was defined as an employee who has

subordinates, oversees staff, is responsible for staff recruitment and training, and holds budgetary accountabilities. Therefore, all levels of managers including Boards of managers were included in this review with the exception of clinical frontline employees, e.g. doctors or nurses, who may have taken on further managerial responsibilities alongside their work but do not have a primary official role as a manager. Those who have specifically taken on a role for quality of care, e.g. the modern matron, were also excluded. Distinction between senior, middle and frontline management was as follows: senior management holds trust-wide responsibilities[26]; middle managers are in the middle of the organisational hierarchy chart and have one or more managers reporting to them[27]; frontline managers are defined as managers at the first level of the organisational hierarchy chart who have frontline employees reporting to them. Board managers include all members of the Board. Although there are overlaps between senior managers and Boards (e.g. chief executive officers (CEOs) may sit on hospital Boards), we aim to present senior and Board level managers separately due to the differences in their responsibilities and position. Only managers who would manage within or govern hospitals were included, with the exclusion of settings that solely served mental health or that comprised solely of non-acute care community services (in order to keep the sample more homogenous). The definition of 'role' focused on actual engagement, time spent and activities that do or did occur rather than those recommended that should or could occur.

### Search strategy
Literature was reviewed between 1 January 1983 and 1 November 2010. Eligible articles were those that described or tested managerial roles pertaining to quality and safety in the hospital setting. Part of the search strategy was based on guidance by Tanon *et al*.[28] EMBASE, MEDLINE, Health Management Information Consortium (HMIC) and PSYCHINFO databases were searched. The search strategy involved three facets (management, quality and hospital setting) and five steps. A facet (i.e. a conceptual grouping of related search terms) for role was not included in the search strategy, as it would have significantly reduced the sensitivity of the search.

Multiple iterations and combinations of all search terms were tested to achieve the best level of specificity and sensitivity. In addition to the key terms, Medical Subject Headings (MeSH) terms were used, which were 'exploded' to include all MeSH subheadings. All databases required slightly different MeSH terms (named Emtree in EMBASE), therefore four variations of the search strategies were used (see online supplementary appendix 1 for the search strategies). Additional limits placed on the search strategy restricted study participants to human and the language to English. The search strategy identified 15 447 articles after duplicates had been removed.

## Screening

Three reviewers (AP, AR and Dina Grishin) independently screened the titles and abstracts of the articles for studies that fit the inclusion criteria. One reviewer (AP) screened all 15 447 articles, while two additional reviewers screened 30% of the total sample retrieved from the search strategy: AR screened 20% and DG screened 10%. On testing inter-rater reliability, Cohen's κ correlations showed low agreement between AR and AP (κ=0.157, p<0.01) and between DG and AP (κ=0.137, p<0.00).[29] However, there was a high percentage of agreement between raters (95% and 89%, respectively), which reveals a good inter-rater reliability.[30] [31] Discrepancies were resolved by discussion and consensus. The main inclusion criteria were that: the setting was hospitals; the population sample reported on was managers; the context was quality and safety; the aim was to identify the managerial activities/time/engagement in quality and safety. The full inclusion/exclusion criteria and screening tool can be accessed in the online supplementary appendices 2–3. Figure 1 presents the numbers of articles included and excluded at each stage of the review process.

Four hundred and twenty-three articles remained for full text screening. One reviewer (AP) screened all articles and a second reviewer (AR) reviewed 7% of these. A moderate agreement inter-rater reliability score was calculated (κ=0.615, p<0.001) with 73% agreement. The primary reoccurring difference in agreement was regarding whether the article pertained to quality of care, owing to the broad nature of the definition. Each article was discussed individually until a consensus was reached on whether to include or exclude. Hand searching and cross-referencing were carried out in case articles were missed by the search strategy or from restriction of databases. One additional article was identified from hand searching,[32] totalling 19 articles included in the systematic review (figure 1).

## Data extraction and methodological quality

The characteristics and summary findings of the 19 included studies are presented in table 1. This table is a simplified version of a standardised template that was used to ensure consistency in data extracted from each article. Each study was assessed using a quality appraisal tool developed by Kmet et al,[34] which comprised of two checklists (qualitative and quantitative). Random included articles (32%) were scored by Ana Wheelock for scoring consistency. All articles were scored on up to 24 questions with a score between 0 and 2; table 2 shows an example definition of what constitutes 'Yes' (2), 'Partial' (1) and 'No' (0) rating criteria. The total percentage scores for each study are presented in table 1. All studies were included regardless of their quality scores. Some cumulative evidence bias may results from two larger data sets split into more than one study each.[35–38] Through a narrative synthesis, we aimed to maintain the original meanings, interpretations and raw data offered by the articles.[39]

## RESULTS

This section provides an overview description of the reviewed studies and their key findings. The findings are considered under four main headings: managerial time spent on quality and safety; managerial quality and safety activities; managerial impact on quality and safety; and contextual factors related to managers' quality and safety role. The section ends with a proposed model to summarise the review findings.

## Description of the studies

From the 19 included studies, the majority were carried out and set in the USA (14 studies) and investigated senior management and/or Boards (13 studies). Of these, 3 focused on senior managers alone (e.g. chief nursing officers), 9 concentrated on Board managers and 1 included a mixture of managerial levels. Only 3 investigated middle managers and 3 examined frontline staff (e.g. clinical directorate managers and unit nurse managers). The settings of the study were mostly trust or hospital-wide; a few articles were set in specific settings or contexts: elderly care,[40] evidence-based medicine,[41] staff productivity,[42] clinical risk management[43] and hospital-acquired infection prevention.[44] Two studies involved specific interventions,[45 46] and 7 studies concentrated specifically on QI rather than quality and safety oversight or routine.[35 40 45–49] There were a mixture of 6 qualitative designs (interviews or focus groups); 8 quantitative survey designs and 5 mix-methods designs. All but one study employed a cross-sectional design.[46] The primary outcome measure used in most studies was perceptions of managerial quality and safety practices. All reported participant perceptions and a majority presented self-reports, i.e. either a mixture of self-reports and peer reports, or self-reports alone.[41 43 45 46] Several studies asked participants about their own and/or other managers' involvement with regard to their specific QI intervention or quality/safety issue.[40 41 44–47] With some variations, the most common research design was to interview or survey senior manager/Board members (particularly Board chairs, presidents and CEOs) perceptions on the Board/senior managers' functions, practices, priorities, agenda, time spent, engagement, challenges/issues, drivers and literacy (e.g. familiarity of key reports) on quality and safety.[35–38 48–51] Five of these studies included objective process/outcome measures, such as adjusted mortality rates.[35 37 38 49 50] No other studies included clinical outcome measures.

The quality assessment scores ranged between 50% and 100%; one study scored (what we consider to be) very low (i.e. <55%), eight studies scored highly (i.e. >75%), two other articles scored highly on one out of two of their studies (quantitative/qualitative) and the remaining eight scored a moderate rating in-between. Almost half of the articles did not adequately describe their qualitative

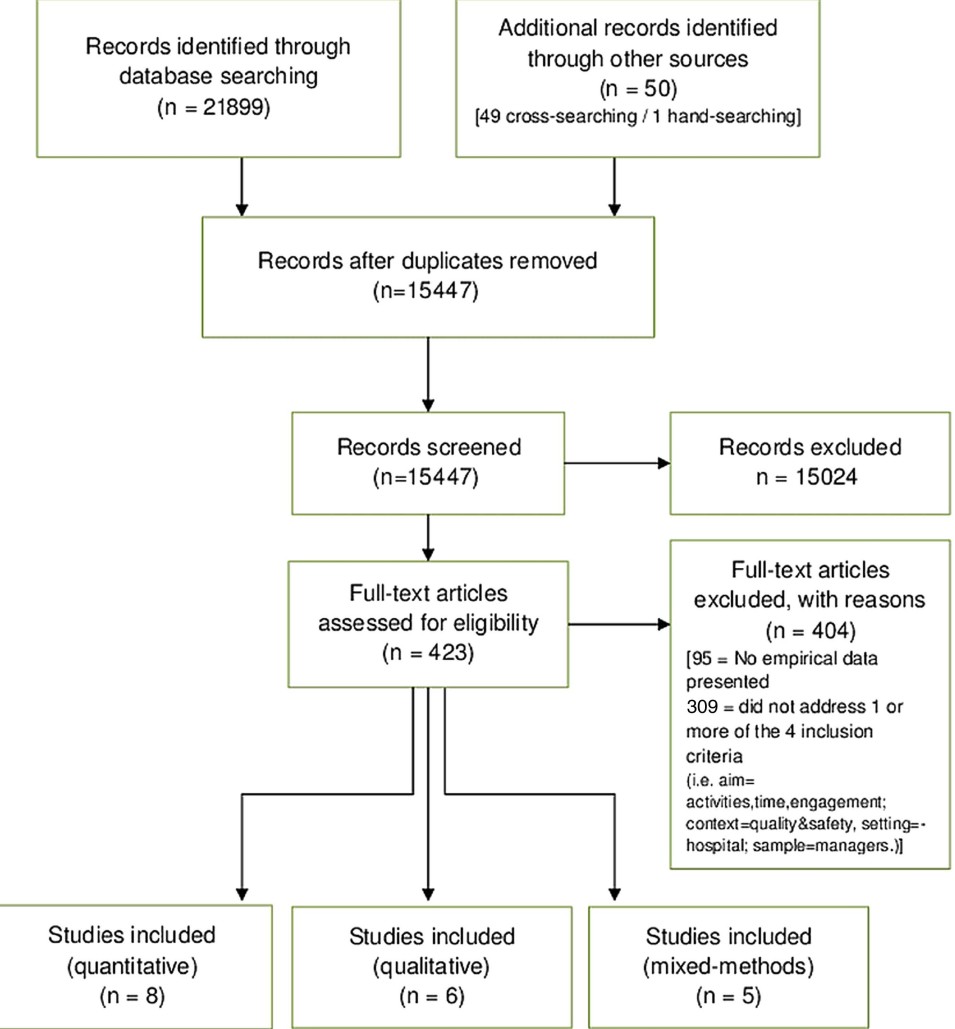

**Figure 1** Review stages based on PRISMA flow diagram.[33]

studies. Specifically, 8 failed to fully describe their qualitative data collection methods, often not mentioning a standardised topic guide, what questions were asked of participants, or no mention of consent and confidentiality assurances. In 7 studies there was no or vague qualitative data analysis description, including omitting the type of qualitative analysis used. Six of the studies showed no or poor use of verification procedures to establish credibility and 9 reported no or poor reflexivity. Positively, all study designs were evident, the context of studies were clear and the authors showed a connection to a wider body of knowledge.

Similarly to the qualitative studies, 7 quantitative studies did not fully describe, justify or use appropriate analysis methods. However, compared with the qualitative studies, the quantitative studies suffered more from sampling issues. Three studies had particularly small samples (e.g. n=35) and one had an especially low response rate of 15%. Participant characteristics were insufficiently described in 5 studies; in one case the authors did not state the number of hospitals included in data analysis. Several studies had obtained ordinal data but only presented percentages, and only one study reported to appropriately controlled for confounding variables. Across all articles, all but 3 studies reported clear objectives and asserted conclusions clearly supported by the data.

### Managerial time spent on quality and safety

The studies on Board level managers highlight an inadequate prioritisation of quality and patient safety on the Board agenda and subsequent time spent at Board meetings. Not all hospitals consistently have quality on their Board agenda, e.g. CEOs and chairpersons across 30 organisations reported that approximately a third of all Board meetings had quality on their agenda,[35] and necessary quality items were not consistently and sometimes never addressed.[36] In all studies examining time spent on quality and safety by the Board, less than half of the total time was spent on quality and safety,[32 37 38 48–51] with a majority of Boards spending 25% or less on quality.[32 38 45 49–51] Findings imply that this may be too low to have a positive influence on quality and safety, as higher quality performance was demonstrated by Boards that spent

above 20%/25% on quality.[49][50] Board members recognised that the usual time spent is insufficient.[48] However, few reported financial goals as more important than quality and safety goals,[32] and health system Boards only spent slightly more time on financial issues than quality.[51] Frontline managers also placed less importance and time on QI,[42] identified as the least discussed topic by clinical managers.[52]

### Managerial quality and safety activities

A broad range of quality-related activities were identified to be undertaken by managers. These are presented by the following three groupings: strategy-centred; data-centred and culture-centred.

### Strategy-centred

Board priority setting and planning strategies aligned with quality and safety goals were identified as Board managerial actions carried out in several studies. High percentages (over 80% in two studies) of Boards had formally established strategic goals for quality with specific targets and aimed to create a quality plan integral to their broader strategic agenda.[32][37] Contrary findings however suggest that the Board rarely set the agenda for the discussion on quality,[37] did not provide the ideas for their strategies[32] and were largely uninvolved in strategic planning for QI.[48] In the latter case, the non-clinical Board managers felt that they held 'passive' roles in quality decisions. This is important considering evidence that connects the activity of setting the hospital quality agenda with better performance in process of care and mortality.[38] Additionally, Boards that established goals in four areas of quality and publicly disseminated strategic goals and reported quality information were linked to high hospital performance.[35][38][50]

### Culture-centred

Activities aimed at enhancing patient safety/QI culture emerged from several studies across organisational tiers.[44][47][48][53] Board and senior management's activities included encouraging an organisational culture of QI on norms regarding interdepartmental/multidisciplinary collaboration and advocating QI efforts to clinicians and fellow senior managers, providing powerful messages of safety commitment and influencing the organisation's patient safety mission.[47][53] Managers at differing levels focused on cultivating a culture of clinical excellence and articulating the organisational culture to staff.[44] Factors to motivate/engage middle and senior management in QI included senior management commitment, provision of resources and managerial role accountability.[40][46] Findings revealed connections between senior management and Board priorities and values with hospital performance and on middle management quality-related activities. Ensuring capacity for high-quality standards also appears within the remit of management, as physician credentialing was identified as a Board managers' responsibility in more than one study.[38][48] From this review it is unclear to what degree Board involvement in the credentialing process has a significant impact on quality.[38][41]

### Data-centred

Information on quality and safety is continually supplied to the Board.[51] At all levels of management, activities around quality and safety data or information were recognised in 6 studies.[35][38][43][45][47][53] Activities included collecting and collating information,[43] reviewing quality information,[35][38][53] using measures such as incident reports and infection rates to forge changes,[53] using patient satisfaction surveys,[35] taking corrective action based on adverse incidents or trends emphasised at Board meetings[38] and providing feedback.[43][47] The studies do not specify the changes made based on the data-related activities by senior managers; one study identified that frontline managers predominantly used data from an incident reporting tool to change policy/practice and training/education and communication between care providers.[45] However, overseeing data generally was found to be beneficial, as hospitals that carried out performance monitoring activities had significantly higher scores in process of care and lower mortality rates than hospitals that did not.[38]

### Managerial impact on quality and safety outcomes

We have considered the associations found between specific managerial involvement and its affect on quality and safety. Here, we summarise the impact and importance of their general role. Of the articles that looked at either outcomes of management involvement in quality or at its perceived importance, 6 articles suggested that their role was beneficial to quality and safety performance.[32][35][38][40][49][53] Senior management support and engagement was identified as one of the primary factors associated with good hospital-wide quality outcomes and QI programme success.[35][38][40][49] Conversely, 6 articles suggest that managers' involvement (from the Board, middle and frontline) has little, no or a negative influence on quality and safety.[35][38][41][42][44][49] Practices that showed no significant association with quality measures included Board's participation in physician credentialing.[35][38] Another noted that if other champion leaders are present, management leadership was not deemed necessary.[44] Two articles identified a negative or inhibitory effect on evidence-based practices and staff productivity from frontline and middle managers.[41][42]

### Contextual factors related to managers' quality and safety role

Most of the articles focused on issues that influenced the managers' role or their impact, as opposed to discussing the role of the managers. These provide an insight in to the types of conditions in which a manager can best undertake their role to affect quality and safety. Unfortunately it appears that many of these conditions are not in place.

 

**Table 1** Table of characteristics and summary findings of included studies

| First author; year (country) | Methods | Sample size (number of organisations) | Population sample (level of management reported on (position of managers)) | Outcome measure | Management roles (managerial quality and safety activities, time spent and engagement and key perceived importance and context factors) | Quality assessment score for qualitative studies | Quality assessment score for quantitative studies | Findings pertaining to research questions (*time spent; activities, engagement; impact (including perceived effectiveness); contextual factors*) |
|---|---|---|---|---|---|---|---|---|
| Baker *et al*; 2010 (Canada)[32] | Mixed methods (interviews, case studies, surveys) | n=15 interviews; n=4 Board case studies; n=79 surveys (79 organisations) | Managers (Board management) | Perceptions of managers on management Board practices in quality and safety | ▸ Less than half (43%) of Boards reported that they addressed quality and patient safety issues in all meetings<br>▸ One-third of Boards spend 25% of their time or more on quality and patient safety issues<br>▸ More than 80% of Boards have formally established strategic goals for quality with specific targets, but a majority of Board chairs indicate that their Boards did not provide the ideas for strategic direction or initiatives<br>▸ Board chairs reported a low participation in education on quality and safety: 43% reported that all the Board members participated, 19% stated that more than half participated and 23% said it was less than a quarter of the Board<br>▸ Most Board chairs (87%) reported Board member induction training on responsibilities for quality and safety, although almost a third (30%) reported few or no opportunities for education on this, 42% reported some opportunities and 28% reported many<br>▸ Approximately half (57%) of the Board chairs acknowledged recruitment of individuals that have knowledge, skills and experience in quality and patient safety onto the Board. A Board skills matrix included quality and safety as one of the competency areas<br>▸ Over half (55%) of board chairs rated their board's effectiveness in quality and safety oversight as very/extremely effective and 40% as somewhat effective | 16/20 (80%) | 12/22 (55%) | Time<br>Activities<br>Impact<br>Context |
| Balding; 2005 (Australia)[46] | Mixed methods (action research, surveys and focus groups) | n=35 (1 hospital) | Managers (middle management (nursing managers and allied health managers)) | Self-reported perceptions of managers on their engagement in a QI programme | Five elements deemed essential to middle manager engagement:<br>▸ Senior management commitment and leadership (e.g. senior management provides strategic direction for QI plan)<br>▸ Provision of resources and opportunities for QI education and information dissemination (e.g. basic QI skills provided to all staff)<br>▸ Senior and middle manager role accountability (e.g. senior managers and middle managers agree QI roles and expectations)<br>▸ Middle manager involvement in QI planning (e.g. senior and middle managers plan together)<br>▸ Middle managers own and operate QI programme (e.g. ongoing review and evaluation of the progress of the QI programme by the middle and senior managers) | 14/20 (70%) | 15/22 (68%) | Activities<br>Engagement<br>Impact |
| Bradely *et al*; 2003 (USA)[47] | Qualitative (interviews) | n=45 (8 hospitals) | Clinical staff and senior management (senior management (unspecified)) | Perceptions of roles and activities that comprise senior management's involvement in quality improvement efforts | Five common roles and activities that captured the variation in management involvement in quality improvement efforts:<br>▸ Personal engagement of senior managers<br>▸ Management's relationship with clinical staff<br>▸ Promotion of an organisational culture of quality improvement<br>▸ Support of quality improvement with organisational structures<br>▸ Procurement of organisational resources for quality improvement efforts | 19/20 (95%) | NA | Activities<br>Engagement<br>Impact |
| Bradely *et al*; 2006 (USA)[40] | Mixed methods (surveys and interviews) | n=63 survey respondents (63 hospitals); n=102 interviewees (13 hospitals) | Managers (senior management (chief operating officer, vice president, medical director, CNO, director of volunteers, programme director)) | Perceptions of management-related factors around the HELP programme | ▸ Providing resources for needed staffing or staff training<br>▸ Promoting the programme among the governing Board, physicians who were initially less involved, and other administrators<br>▸ Senior management support reported as the primary enabling factor in the implementation of such programmes (96.6%), along with a lack of support as the primary reason for not implementing the programme (65.0%)<br>▸ The interviews supported that having an administrative champion was considered essential to their programme's success | 19/20 (95%) | 17/22 (77%) | Activities<br>Engagement<br>Impact<br>Context |

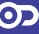

**Table 1** Continued

| First author; year (country) | Methods | Sample size (number of organisations) | Population sample (level of management reported on (position of managers)) | Outcome measure | Management roles (managerial quality and safety activities, time spent and engagement and key perceived importance and context factors) | Quality assessment score for qualitative studies | Quality assessment score for quantitative studies | Findings pertaining to research questions (*time spent; activities, engagement; impact (including perceived effectiveness); contextual factors*) |
|---|---|---|---|---|---|---|---|---|
| Braithwaite *et al*; 2004 (Australia)[52] | Mixed methods (ethnographic work, observations and focus groups) | n=64 managers in focus groups (1 hospital); ethnographic case studies and n=4 observed (2 hospitals) | Managers (frontline management (medical managers, nurse managers and allied health managers)) | Observations and self-reported perceptions of clinician-managers' activities | ▶ Quality was the least discussed topic (e.g. continuous quality improvement)<br>▶ The most discussed topic was people (e.g. staffing, delegating) and organisational issues, e.g. beds and equipment | 16/20 (80%) | NA | Time |
| Caine and Kenwrick; 1997(UK)[41] | Qualitative (interviews) | n=10 (2 hospitals) | Managers (middle management (clinical directorate managers)) | Self-reported perceptions of managers on the managers' role in facilitating evidence-based practice in their nursing teams | ▶ Managers saw their role in research implementation as a facilitator, ensuring quality and financial objectives and standards were met<br>▶ Managers perceived their facilitatory behaviours produced a low level of clinical change<br>▶ Managers are not actively advocating research-based practice and failing to integrate it into everyday practice. Their behaviour inhibited the development of evidence-based nursing practice<br>▶ Devolved responsibility of use of research to individual professionals | 14/20 (70%) | NA | Activities<br>Impact |
| Fox, Fox and Wells; 1999 (USA)[42] | Quantitative (surveys and self kept activity logs) | n=16 (1 hospital) | Managers (frontline management (nurse administrative managers (NAMs))) | Self-reported perceptions of managers on their activities impacting unit personnel productivity and monitored time/effort allocated to each function and managers' hours worked, patient admissions and length of stay | ▶ The small amount of total management allocated to QI (2.6%) was the least time spent of all management functions<br>▶ A negative relationship between time spent in QI activities and unit personnel productivity. An increase (from 2.5% to 5%) in QI time/effort by NAMs would reduce staff productivity significantly by approximately 8%<br>▶ The greater the experiences of NAMs as managers, the more time spent on QI. These seasoned NAMs spent more time on monitoring, reporting QI results and quality improvement teams (statistics nor provided) | NA | 13/22 (59%) | Time<br>Activities<br>Impact (objective outcome measure) |
| Harris; 2000 (UK)[43] | Quantitative (surveys) | n=42 (42 hospitals) | Managers (middle management (nurse managers)) | Self-reported perceptions of managers on managers' quality and safety practices | ▶ The majority of managers (91%) who received collated incident information used it to feed back to their own staff. 60% always fed back to staff, 28% sometimes did, 2% never did<br>▶ Of the trusts that had written guidance on types of clinical incident to report, 80% of managers had general guidance and fewer (20%) had written specialty specific guidance<br>▶ 76% of managers reported information collation of clinical incidents. Of these, 59% were involved in data collection themselves | NA | 13/22 (59%) | Activities |
| Jha and Epstein; 2010 (USA)[50] | Quantitative (surveys) | n=722 (767 hospitals) | Managers (Board) | Perceptions of managers on the role of managers in quality and safety and quality outcome measurement (from HQA) i.e. 19 practices for care in 3 clinical conditions | ▶ Two-thirds (63%) of Boards had quality as an agenda item at every meeting<br>▶ Fewer than half (42%) of the hospitals spent at least 20% of the Board's time on clinical quality<br>▶ 72% of Boards regularly reviewed a quality dashboard<br>▶ Most respondents reported that their Boards had established, endorsed or approved goals in four areas of quality: hospital-acquired infections (82%), medication errors (83%), the HQA/Joint commission core measures (72%), and patient satisfaction (91%)High-performing hospitals were more likely than low-performing hospitals to have:<br>▶ Board reviews of a quality dashboard regularly (<0.001) and of clinical measures (all <0.05) | NA | 22/22 (100%) | Time<br>Activities<br>Impact (objective outcome measure)<br>Context |

Continued

**Table 1** Continued

| First author; year (country) | Methods | Sample size (number of organisations) | Population sample (level of management reported on (position of managers)) | Outcome measure | Management roles (managerial quality and safety activities, time spent and engagement and key perceived importance and context factors) | Quality assessment score for qualitative studies | Quality assessment score for quantitative studies | Findings pertaining to research questions (*time spent; activities, engagement; impact (including perceived effectiveness); contextual factors*) |
|---|---|---|---|---|---|---|---|---|
| Jiang et al; 2008 (USA)[37] | Quantitative (surveys) | n=562 (387 hospitals) | Managers (Board and senior management (presidents/CEOs)) | Perceptions of managers on managers' practices in quality and safety; and outcomes of care (composite scores of risk-adjusted M indicators) | ▸ Quality performance on the agenda at every Board meeting (0.003)<br>▸ At least 20% of Board time on clinical quality (0.001)<br>▸ Has a quality subcommittee (0.001)<br>▸ 75% of CEOs reported that most to all of the Board meetings have a specific agenda item devoted to quality. Only 41% indicated that the Boards spend more than 20% of its meeting time on the specific item of quality. The following activities were most reported to be performed:<br>▸ Board establishing strategic goals for QI (81.3%)<br>▸ Use quality dashboards to track performance (86%)<br>▸ Follow-up corrective actions related to adverse events (83%) The following activities were least reported to be performed:<br>▸ Board involvement in setting the agenda for the discussion on quality (42.4%)<br>▸ Inclusion of the quality measures in the CEO's performance evaluation (54.6%)<br>▸ Improvement of quality literacy of Board members (48.9%)<br>▸ Board written policy on quality and formally communicated it (30.8%) | NA | 20/26 (77%) | Time<br>Activities<br>Impact (objective outcome measure)<br>Context |
| Jiang et al; 2009 (USA)[38] | Quantitative (surveys) | n=490 (490 hospitals) | Managers (Board and senior management (CEOS and hospital presidents reports)) | Perceptions of managers on manager's practices in quality and safety; and POC measures (20 measures in 4 clinical areas); and outcome measures (composite scores of risk-adjusted M indicators) | Board practices found to be associated with better performance (all p<0.05) in POC and adjusted M included:<br>▸ Having a Board quality committee (83.8%POC, 6.2M versus 80.2%POC, 7.9M without a committee)<br>▸ Establishing strategic goals for quality improvement (82.8% POC, 6.6M versus 80.3% POC, 7.9M)<br>▸ Being involved in setting the quality agenda for the hospital (83.2% POC, 6.4M versus 80.9% POC, 7.7M)<br>▸ Including a specific item on quality in Board meetings (83.2% POC, 6.5M versus 78.5% POC, 8.6M)<br>▸ Using a dashboard with national benchmarks and internal data that includes indicators for clinical quality, patient safety and patient satisfaction (all above 80% POC and below 6.5M versus all above 80%POC and above 7M)<br>▸ Linking senior executives' performance evaluation to quality and patient safety indicators (83.1% POC, 6.6M versus 80.4% POC, 7.6M) Practices that did NOT show significant association with the quality measures for process and M include:<br>▸ Reporting to the Board of any corrective action related to adverse events (82.5% POC, 7.0M versus 81.8% POC, 6.6M)<br>▸ Board's participation in physician credentialing (82.8% POC, 6.9M versus 81.5% POC, 6.9M)<br>▸ Orientation for new Board members on quality(82.9% POC, 6.8M versus 81.7% POC, 7.0M)<br>▸ Education of Board members on quality issues (82.8% POC, 7.0M versus 81.9% POC, 6.9M) | NA | 22/24 (92%) | Activities<br>Impact (objective outcome measure)<br>Context |
| Joshi and Hines; 2006 (USA)[35] | Mixed methods (surveys and interviews) | n=37 survey respondents; n=47 interviewees (30 hospitals) | Managers (Board and senior management (CEOs, Board chairs)) | Perceptions of managers on managers' practices in quality and safety and ACM and risk-adjusted M. | ▸ Board engagement in quality was reported as satisfactory (7.58 by CEOs and 8.10 by Chairs on a 1–10 scale where 10 indicates greatest satisfaction)<br>▸ Board engagement was positively associated with perceptions of the rate of progress in improvement (r=0.44, p =0.05), and marginally associated with ACM scores (r=0.41, p=0.07)<br>▸ Approximately one-third of Board meetings are devoted to discussing quality issues (reported at 35% by CEOs and 27% by Chairs) | 12/20 (60%) | 16/20 (80%) | Time<br>Activities<br>Engagement<br>Impact (objective outcome measure)<br>Context |

Continued

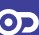

**Table 1** Continued

| First author; year (country) | Methods | Sample size (number of organisations) | Population sample (level of management reported on (position of managers)) | Outcome measure | Management roles (managerial quality and safety activities, time spent and engagement and key perceived importance and context factors) | Quality assessment score for qualitative studies | Quality assessment score for quantitative studies | Findings pertaining to research questions (*time spent; activities, engagement; impact (including perceived effectiveness); contextual factors*) |
|---|---|---|---|---|---|---|---|---|
| | | | | | ▶ Integrating Quality Planning and Strategic Planning was also rated as satisfactory (7.67 by CEOs and 8.85 by Chairs) | | | |
| | | | | | ▶ Approximately two-thirds of respondents reported using patient satisfaction surveys (70% and 65% reported by CEOs and Chairs, respectively) | | | |
| | | | | | ▶ Low level of CEO expertise in QI, as reported by themselves (2.70) and by Board Chairs (3.35%) on a scale of 1–10 where 1 is low familiarity and 10 is high familiarity | | | |
| Levey et al; 2007 (USA)[48] | Qualitative (interviews) | n=96 (18 hospitals) | Managers (Board and senior management (hospital Board members, CEOs, chief medical officers, chief quality officers, medical staff leaders)) | Perceptions of managers' role in quality and safety | ▶ Few CEOs were willing to take the lead for transformation to a 'culture of quality' | 13/20 (65%) | NA | Time Activities Engagement Context |
| | | | | | ▶ Board members were largely uninvolved in strategic planning for QI | | | |
| | | | | | ▶ In terms of the Board's quality functions, respondents largely agreed that physician credentialing was their critical responsibility | | | |
| | | | | | ▶ Non-physicians reported that they felt relegated to 'passive' roles in decisions on quality and seemed reluctant to assume leadership roles in the quality domain | | | |
| | | | | | ▶ Board meeting agendas maintained a focus on financial issues, although patient safety/care and QI were gaining prominence | | | |
| | | | | | ▶ About half of the respondents said that quality was not sufficiently highlighted during meetings. Estimates of time devoted to quality and safety issues at Board meetings were between 10% and 35% | | | |
| Mastal, Joshi and Shulke; 2007 (USA)[36] | Qualitative (interviews and a focus group) | n=73 interviewees; 1 focus group (63 hospitals) | Managers (Board and senior management (Board chairs, CEOs, CNOs)) | Perceptions of managers on managers' role in quality and safety | ▶ Two CNOs reported that nursing quality was never addressed at Board meetings | 12/20 (60%) | NA | Time Context |
| | | | | | ▶ Few of the CNOs, CEOs and Board chairs responded that issues are discussed more frequently, such as at every meeting | | | |
| | | | | | ▶ Quality and patient safety measures for nurses are not consistently addressed during all hospital Board meetings | | | |
| | | | | | ▶ Staffing concerns are the most frequent measure of nursing quality reported at the Board level | | | |
| Poniatowski, Stanley and Youngberg; 2005 (USA)[45] | Quantitative (surveys) | n=515 (16 academic medical centers) | Managers (frontline management—unclear whether frontline or middle managers (unit nurse managers)) | Self-reported perceptions of managers on their practices with PSN | ▶ Managers reviewed on average 65% of the PSN events reportedAs a result of what was learned from PSN data, 162 managers detailed their changes made to: | NA | 10/20 (50%) | Activities |
| | | | | | ▶ Policies and practices (59%) | | | |
| | | | | | ▶ Training, education and communication between care providers (27%) | | | |
| | | | | | ▶ Purchase of new equipment and supplies (8%) | | | |
| | | | | | ▶ Staffing (6%) | | | |
| Prybil et al; 2010 (USA)[51] | Quantitative (surveys) | n=123 (712 hospitals) | Managers (Board and senior management (CEOs and Boards)) | Perceptions of managers on their role in quality and safety | ▶ Health system Boards spent 23% of their Board meeting time on quality and safety issues. They only spent slightly more on financial issues (25.2%) and strategic planning (27.2%) | NA | 14/22 (64%) | Time Activities Context |
| | | | | | ▶ Almost all (96%) CEOs said that the Boards regularly received formal written reports on quality targets | | | |
| | | | | | ▶ 88% of CEOs said that the Boards had assigned quality and safety oversight to a standing Board committee | | | |
| | | | | | ▶ All but one (98.9%) of the CEOs stated that they have specific performance expectations and criteria related to quality and safety | | | |
| | | | | | ▶ CEOs reported 59% of the Boards formally adopted system-wide measures and standards for quality | | | |

Continued

**Table 1** Continued

| First author; year (country) | Methods | Sample size (number of organisations) | Population sample (level of management reported on (position of managers)) | Outcome measure | Management roles (managerial quality and safety activities, time spent and engagement and key perceived importance and context factors) | Quality assessment score for qualitative studies | Quality assessment score for quantitative studies | Findings pertaining to research questions (*time spent*; *activities, engagement*; *impact (including perceived effectiveness)*; *contextual factors*) |
|---|---|---|---|---|---|---|---|---|
| Saint *et al*; 2010 (USA)[44] | Qualitative (interviews) | n=86 (interviewees) (14 hospitals) | Senior hospital staff and managers (mixed levels (nurse managers, chief physicians, Chairs of medicine, chief of staffs, hospital directors, CEOs and clinical non-managerial staff)) | Perceptions of managers on managers' practices in HAI | ▶ Although committed leadership by CEOs can be helpful, it was not always necessary, provided that other hospital leaders were committed to infection prevention Behaviours of leaders who successfully implemented/facilitated practices to prevent HAI: <br> ▶ Cultivated a culture of clinical excellence and kept their eye on improving patient care <br> ▶ Developed a vision <br> ▶ Articulated the organisational culture well and conveyed that to staff at all levels <br> ▶ Focused on overcoming barriers and dealing directly with resistant staff or process issues that impeded prevention of HAI <br> ▶ Cultivated leadership skills and inspired the people they supervised (motivating and energising them to work towards the goal of preventing HAI) <br> ▶ Thought strategically while acting locally; planned ahead and left few things to chance <br> ▶ They did the politicking before issues arose for committee votes <br> ▶ They leveraged personal prestige to move initiatives forward <br> ▶ They worked well across disciplines | 16/20 (80%) | NA | Activities Engagement Impact |
| Vaughn *et al*; 2006 (USA)[49] | Quantitative (surveys) | n=413 (413 hospitals) | Managers (Board and senior management (chief executives and senior quality executives; Board, executives, clinical leadership)) | Perceptions of managers on managers' role in QI and observed hospital quality index outcomes (risk-adjusted measures of morbidity, M and medical complications) | ▶ 72% of hospital Boards spent one-quarter of their time or less on quality-of-care issues. About 5% of Boards spent more than half of their time on these issues <br> ▶ A majority of respondents reported great influence from government and regulatory agencies (87%), consumers (72%) and accrediting bodies (74%) on quality priorities. Although 44% of respondents also noted that multiple government and regulatory requirements were unhelpfulBetter QIS are associated with hospitals where the Board: <br> ▶ Spends more than 25% of their time on quality issues (QIS 83–QIS mean 100 across hospitals) <br> ▶ Receives a formal quality performance measurement report (QIS 302) <br> ▶ Bases the senior executives' compensation in part on QI performance (QIS 239) <br> ▶ Engages in a great amount of interaction with the medical staff on quality strategy | NA | 21/22 (95%) | Time Activities Engagement Impact (objective outcome measure) Context |
| Weingart and Page; 2004 (USA)[53] | Qualitative (case study documentation analysis and meeting discussions and focus group) | n=30 (10 hospitals and other stakeholder organisations) | Managers (senior management (executives)) | Perceptions of managers on manager's practices in quality and safety | Executives developed and tested a set of governance best practices in patient safety, such as: <br> ▶ Creation of a Board committee with explicit responsibility for patient safety <br> ▶ Development of Board level safety reports, introduction of educational activities for Board members <br> ▶ Participation of Board members in executive walk rounds <br> ▶ Executives reviewed measures to assess safety (e.g. incident reports, infection rates, pharmacist interventions, readmissions, etc) <br> ▶ Executives endorsed a statement of public commitment to patient safety <br> ▶ Executives believed their behaviours affected their organisations' patient safety mission | 14/20 (70%) | NA | Activities Impact |

ACM, appropriate care measure; CEO, chief executive officer; CNO, chief nursing officer; HAI, healthcare-associated infection; HQA, Hospital Quality Alliance; M, mortality; NA, not applicable; POC, process of care; PSN, Patient Safety Net; QI, quality improvement; QIS, quality index scores.

| Table 2 | Example of rating criteria from Kmet's quality assessment tool[34] |
|---|---|
| Rating | Criteria to verify whether question or objective is sufficiently described |
| Yes | Is easily identified in the introductory section (or first paragraph of methods section). Specifies (where applicable, depending on study design) *all* of the following: purpose, participants/target population, and the *specific* intervention(s)/association(s)/descriptive parameter(s) under investigation. A study purpose that only becomes apparent after studying other parts of the paper is *not* considered sufficiently described |
| Partial | Vaguely/incompletely reported (e.g. "describe the effect of" or "examine the role of" or "assess opinion on many issues" or "explore the general attitudes"...); *or* some information has to be gathered from parts of the paper other than the introduction/background/objective section |
| No | Question or objective is not reported, or is incomprehensible |
| N/A | Should not be checked for this question |

Two studies found that a Board quality committee is a positive variable in quality performance, but that fewer than 60% had them.[38 50] Similarly, compensation and performance evaluation linked to executive quality performance was identified in 4 articles [35 37 38 49] and associated with better quality performance indicators,[38 49] but quality measures were insufficiently included in CEOs' performance evaluation.[35 37] The use of the right measures to drive QI was raised in relation to Board managerial engagement in quality [35] and to impact on patient care improvement,[51] yet almost half of this sample did not formally adopt system-wide measures and standards for quality. To aid them in these tasks, evidence indicates the common use of QI measure tools, such as a dashboard or scorecard,[37 49 50] with promising associations between dashboard use and quality outcomes.[38 50]

Other factors linked to quality outcomes include management–staff relationship/high interactions between the Board and medical staff when setting quality strategy,[49] and managerial expertise. Although a connection between knowledge and quality outcomes was not found,[38] high performing hospitals have shown higher self-perceived ability to influence care, expertise at the Board and participation in training programmes that have a quality component.[50] Disappointingly, there is a low level of CEO knowledge on quality and safety reports,[35] possibly little Boardroom awareness on salient nursing quality issues,[36] and little practice identified to improve quality literacy for the Board.[32 37] There is however promise for new managers through relevant training at induction and by recruitment of those with relevant expertise.[32]

### The quality management IPO model

The input process output (IPO) model is a conceptual framework that helps to structure the review findings in a useful way (see figure 2).[54 55] This literature may be conceptualised by considering what factors contribute (input) to managerial activities (process) that impact on quality and safety (output). The three factors are inter-related and interchangeable, presented by the cyclical interconnecting diagram. This diagram enables a clearer mental picture of what a manager should consider for their role in quality and safety. Specifically, the input

factors suggest certain organisational factors that should be put in place alongside individual factors to prepare for such a role (e.g. standardised quality measures, motivation, education and expertise, and a good relationship with clinicians). The processes present the strategy, culture and data-centred areas where managers (according to the literature) are and/or should be involved (e.g. driving improvement culture, goal setting and providing feedback on corrective actions for adverse events). The outputs identify managerial influences that are positive, negative or have little or no established association with quality performance (e.g. positive outcomes of care, achieving objectives and engaging others in quality of care). This helps to identify areas where it is possible to make an impact through the processes mentioned. With further empirical studies on this topic, this model could be strengthened to become a more robust set of evidence-based criteria and outcomes.

### DISCUSSION

Our review examined the role of managers in maintaining and promoting safe, quality care. The existing studies detail the time spent, activities and engagement of hospital managers and Boards, and suggest that these can positively influence quality and safety performance. They further reveal that such involvement is often absent, as are certain conditions that may help them in their work.

Evidence from the review promotes hospitals to have a Board quality committee, with a specific item on quality at the Board meeting, a quality performance measurement report and a dashboard with national quality and safety benchmarks along with standardised quality and safety measures. Outside of the Boardroom, the implications are for senior managers to build a good infrastructure for staff–manager interactions on quality strategies and attach compensation and performance evaluation to quality and safety achievements. For QI programmes, managers should keep in mind its consistency with the hospital's mission and provide commitment, resources, education and role accountability. Literature elsewhere supports much of these findings, such as the use of quality measurement tools[21 56] better quality-associated

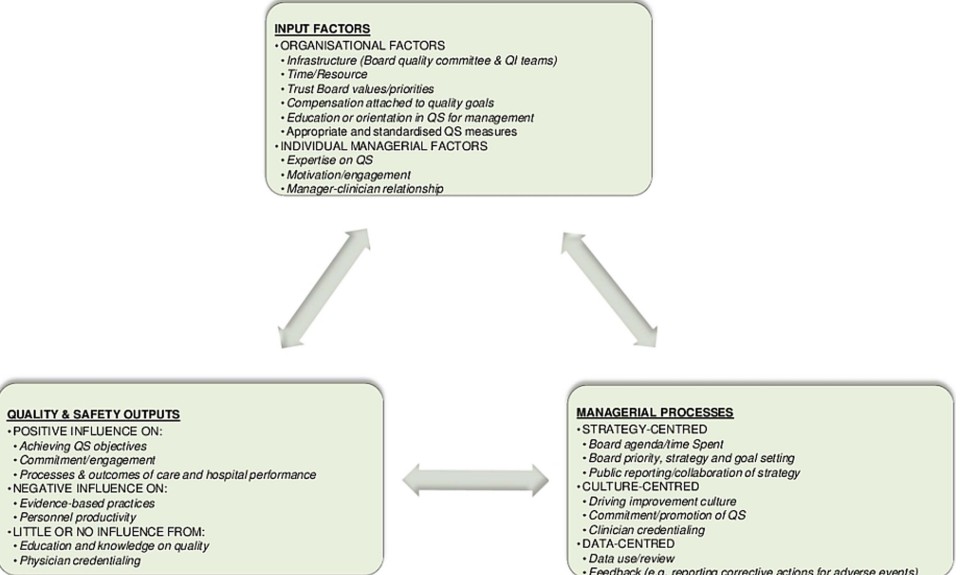

**Figure 2** The quality management IPO model (IPO, input process output; QI, quality improvement).

compensation, a separate quality committee,[16 57] and has also emphasised poor manager–clinician relationships as damaging to patients and QI.[58 59]

Some of the variables that were shown to be associated with good quality performance, such as having a Board committee, compensation/performance and adoption of system-wide measures, were lacking within the study hospitals. There are also indications of the need to develop Board and senior managerial knowledge and training on quality and safety. Furthermore, this review indicates that many managers do not spend sufficient time on quality and safety. The included studies suggest time spent by the Board should exceed 20–25%, yet the findings expose that certain Boards devote less time than this. Inadequacies of time allocated to quality at the Board meeting hold concerning implications for quality. If little time is taken to consider quality of care matters at the highest level, an inference is that less attention will be paid to prevention and improvement of quality within the hospital. While the position that the item appears on the agenda is deemed of high importance, it is unimportant if the duration on this item is overly brief. In this vein, the inadequate time on quality spent by some may reflect their prioritisation on quality in relation to other matters discussed at the meetings or the value perceived to be gained from discussing it further. It might instead however be indicative of the difficulties in measuring time spent on quality by management. Some of these studies provide us not necessarily with Board managers' time on quality and safety but their time spent on this at Board meetings. The two may not equate and time spent on quality may not necessarily be well spent.[36] The emerging inference that managers greatly prioritise other work over quality and safety is not explicit, with further research required to identify what time is actually devoted and required from managers inside and outside of the Boardroom. Perhaps encouragingly, the more

recent studies present more time spent on quality and safety than the earlier studies. Yet even the most recent empirical studies not included in our review conclude that much improvement is required.[60]

This review presents a wide range of managerial activities, such as public reporting of quality strategies and driving an improvement culture. It further highlights the activities that appear to affect quality performance. Priorities for Boards/managers are to engage in quality, establish goals and strategy to improve care, and get involved in setting the quality agenda, support and promote a safety and QI culture, cultivate leaders, manage resisters, plan ahead and procure organisational resources for quality. Again, much of the findings support the assertions made in the non-empirical literature. Above all, involvement through action, engagement and commitment has been suggested to positively affect quality and safety.[61] While researchers have stressed the limited empirical evidence showing conclusive connection between management commitment and quality,[21] some supporting evidence however can be unearthed in research that concentrates on organisational factors related to changes made to improve quality and safety in healthcare.[62–64] In addition to this evidence, a few studies have specifically investigated the impact that hospital managers have on quality and safety (rather than examination of their role). These studies have shown senior managerial leadership to be associated with a higher degree of QI implementation,[65] promotion of clinical involvement,[66 67] safety climate attitudes[68] and increased Board leadership for quality.[57] A clear case for the positive influence of management involvement with quality is emerging both from the findings of our review and related literature.

There is a dearth of empirical research on the role of hospital managers in quality of care and patient safety and QI. This evidence is further weakened by the largely

descriptive nature of many of the studies. They mostly lack theoretical underpinnings and appropriate objective measures. Very few studies reported objective clinical quality outcome measures that better show the influence of managerial actions. Moreover, the content of many of the articles was dominated by the contextual issues surrounding managers' roles, rather than actual manager practices. Some of the outlined managerial actions would further benefit from more detail, e.g. the literature fails to present changes made based on the data-related activities at the Board or senior management level. Only one study clearly demonstrated that senior management and Board priorities can impact on middle management quality-related activities and engagement. Considering the likely influence that seniors have on their managers, examination of the interactions between the different roles held (e.g. Boards setting policies on quality and middle managers implementing them) would improve our understanding of how these differences reflect in their time spent and actions undertaken. Supplementary work could also resolve contradictions that were found within the review, clarifying for example, the positive impact of managerial expertise versus knowledge on quality and who sets the Board agenda for the discussion on quality. Research on this area is particularly required to examine middle and frontline managers, to take into consideration non-managers' perceptions, and to assess senior managers' time and tasks outside of the Boardroom. Future studies would benefit from better experimental controls, ideally with more than one time point, verifications and reflections on qualitative work, robust statistical analysis, appropriate study controls, consideration of confounding variables, and transparent reporting of population samples, methodologies, and analyses used. Box 1 presents the key messages from this review.

---

> **Box 1**  Key messages from the systematic literature review

> ▶ There is a dearth of empirical evidence on hospital managerial work and its influence on quality of care.
> ▶ There is some evidence that Boards'/managers' time spent, engagement and work can influence quality and safety clinical outcomes, processes and performance.
> ▶ Some variables associated with good quality performance were lacking in study hospitals.
> ▶ Many Board managers do not spend sufficient time on quality and safety.
> ▶ There is a greater focus on the contextual issues surrounding managers' roles than on examining managerial activities.
> ▶ Research is required to examine middle and frontline managers, to take into consideration non-managers' perceptions, and to assess senior managers' time and tasks outside of the Boardroom. More robust methodologies with objective outcome measures would strengthen the evidence.
> ▶ We present a model to summarise the evidence-based promotion of conditions and activities for managers to best affect quality performance.

## Review limitations

There are several limitations of the present review pertaining to the search strategy and review process, the limited sample of studies, publication bias, and limitations of the studies themselves. Specifically, the small number of included studies and their varied study aims, design and population samples make generalisations difficult. Grouped demographics, such as middle management, are justified by the overlap between positions. With more literature on this topic, distinctions could be made between job positions. Furthermore, more research on lower levels of management would have provided a better balanced review of hospital managers' work and contributions to quality. Restricting the language of studies to English in the search strategy is likely to have biased the findings and misrepresent which countries conduct studies on this topic. There is an over-reliance on perceptions across the studies, which ultimately reduces the validity of the conclusions drawn from their findings. As most of the study findings relied on self-reports, social desirability may have resulted in exaggerated processes and inflated outputs. Although, encouragingly, one of the included studies found that managers who perceived their Boards to be effective in quality oversight were from hospitals that had higher processes-of-care scores and lower risk adjusted mortality. The quality assessment scores should be viewed with caution; such scores are subjective and may not take into consideration factors beyond the quality assessment scale used. Owing to the enormity of this review, the publication of this article is some time after the search run date. As there is little evidence published on this topic, we consider this not to greatly impact on the current relevance of the review, particularly as the literature reviewed spans almost three decades. However, we acknowledge the need for an update of the data as a limitation of this review.

## CONCLUSION

The modest literature that exists suggests that managers' time spent, engagement and work can influence quality and safety clinical outcomes, processes and performance. Managerial activities that affect quality performance are especially highlighted by this review, such as establishing goals and strategy to improve care, setting the quality agenda, engaging in quality, promoting a QI culture, managing resisters and procurement of organisational resources for quality. Positive actions to consider include the establishment of a Board quality committee, with a specific item on quality at the Board meeting, a quality performance measurement report and a dashboard with national quality and safety benchmarks, performance evaluation attached to quality and safety, and an infrastructure for staff–manager interactions on quality strategies. However, many of these arrangements were not in place within the study samples. There are also indications of a need for

managers to devote more time to quality and safety. More than one study suggest time spent by the Board should exceed 20–25%, yet the findings expose that certain Boards devote less time than this. Much of the content of the articles focused on such contextual factors rather than on the managerial role itself; more empirical research is required to elucidate managers' actual activities. Research is additionally required to examine middle and frontline managers, non-manager perceptions, and to assess senior managers' time and tasks outside of the Boardroom. We present the quality management IPO model to summarise the evidence-based promotion of conditions and activities in order to guide managers on the approaches taken to influence quality performance. More robust empirical research with objective outcome measures could strengthen this guidance.

**Acknowledgements** The authors would like to thank Miss Dina Grishin for helping to review the abstracts and Miss Ana Wheelock for helping to assess the quality of the articles.

**Contributors** All coauthors contributed to the study design and reviewed drafts of the article. The first author screened all the articles for inclusion in this review and appraised the study quality. AR and Dina Grishin screened a sample of these at title/abstract and full text, and Ana Wheelock scored the quality of a sample of the included articles.

**Funding** This work was supported by funding from the Health Foundation and the National Institute for Health Research (grant number: P04636).

**Competing interests** None.

**Provenance and peer review** Not commissioned; externally peer reviewed.

**Data sharing statement** The extraction table of the included studies and individual study quality scores can be made available on request to the corresponding author at a.parand@imperial.ac.uk.

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
