## [Reviewer comments · BMJ Open]

Some articles will have been accepted based in part or entirely on reviews undertaken for other BMJ Group journals. These will be reproduced where possible.

ARTICLE DETAILS

TITLE (PROVISIONAL)	The Role of Hospital Managers in Quality and Patient Safety. A Systematic Review
AUTHORS	Parand, Anam; Dopson, Sue; Renz, Anna; Vincent, Charles

VERSION 1 - REVIEW

REVIEWER	Deirdre Thornlow Duke University School of Nursing
REVIEW RETURNED	11-Apr-2014

GENERAL COMMENTS	Well written article on a very important topic. In the methods section (screening), I would like to have seen higher percentage of two reviews (i.e., 7% with only 73% agreement seems insufficient). Nice diagram of screening process (Figure 1). I found the conclusion to be contradictory as stated, "Managerial activities that affect quality performance are especially highlighted by this review,...However, studies additionally show that many of these contextual factors and activities are lacking..." Also, several conclusions are based on only 1 or 2 studies.
---

REVIEWER	Kevin Kenward HRET, USA
	Maulik Joshi, my boss, is the author of two of the reviewed articles.
REVIEW RETURNED	25-May-2014

GENERAL COMMENTS	Literature reviews are important due to the ever-increasing output of publications researchers cannot be expected to examine in detail every single new paper relevant to their interests. Thus, it is both advantageous and necessary to rely on regular summaries of the recent literature. The emphasis here is on the word "recent." The authors never give an explanation as to why they reviewed articles only up to the year 2010. That means their most recent article reviewed is four years old by now. Researchers must survey the field and its trends – and this means right up to the moment. For literature reviews to be useful they need to be compiled in a professional way. The researchers seem to have been comprehensive in their search having identified and reviewed over 15,000 articles. The authors also attempt to assess the quality of the
--

	19 articles selected for review by using an assessment tool. However, no explanation is given as to the assessment measures or how to interpret them. It would be helpful to know what a "good" score is since the scores range from 55% to 100%. Table 1 presents a good summary of the 19 articles. In addition to or substitution of this table could be a table that indicates which studies address which of the five research questions. For instance, the studies that relate to the question of time managers spend on time and safety could be listed or presented in summary form. The definition of "manager" seems rather broad. The review would also have benefited from a discussion about the different managerial roles and how those roles relate to quality and safety initiatives. For example, Boards set policies and middle managers implement policies. The different roles and responsibilities would then impact how much time was spent on quality activities. There also could have been more critical analysis of the distinction between authors' interpretation of their data and the actual empirical evidence presented. A good review critically analyses how accurately previous authors have reported their findings and whether they have refrained from asserting conclusions not supported by data. Without going through the summary of the 19 articles it is difficult to determine when a statement is being supported by qualitative or quantitative research. The authors innovative in presenting the theories and results using the input process output (IPO) model. Although the model is presented in one short paragraph and a diagram. There is no discussion as to how to use the model or what insights it provides.
--	--

REVIEWER	Elizabeth West Centre for Positive Ageing University of Greenwich London England
REVIEW RETURNED	14-Jun-2014

GENERAL COMMENTS	This is a really interesting paper and has great potential as a springboard for further research. I would like to raise some points for clarification and qualify some of my answers to the checklist above.  1. Abstract: could you be more specific about what the actual findings of the review were? I would also like to have a more complete summary of the findings at the end of the paper, immediately before the discussion. The discussion does bring in a lot of new contextual material which perhaps should be cited and discussed at the beginning of the paper and could then be used in the discussion section to relate the findings of the review to this literature. 2. This is a review of mainly descriptive studies. The authors have conducted a systematic and extensive search but I would like to see more attention to and critique of the main research designs employed in this body of literature. 3. In the first paragraph of "methods" can you please spell out the definitions of "quality of care" and "patient safety" that you are using so that the reader has them to hand?
---

	4. I am not sure that the definition of “manager” on page 6 is entirely consistent with the description of the study participants in table 1, e.g. inclusion of unit nurse managers. 5. Why did you exclude mental health acute settings? 6. What is the technical meaning of “facets” at the bottom of page 6? 7. Were any articles not included in the study because they were not of sufficiently high quality? 8. What was done about the 27% of articles about which there was disagreement between reviewers (top of page 8)? 9. It would be good to have more discussion of the typical research designs used in this body of literature as well as their overall quality. 10. Summarise the findings and discuss the strength of the evidence. Is it a problem that in table 1, most of the outcomes are “perceptions” rather than patient outcomes? If managers spend a lot of their time on quality issues, would you not expect that they would then report that this has a significant impact on these issues? Is there evidence in this paper for the wider claims emerging from this paper, in, for example, paragraph 2 of the discussion and there on to the end of the paper? Because I think that this is a potentially important paper, I have highlighted issues of research design in the review. I think it is important that the reader knows more about the kind of studies that are included; this then informs how you discuss the strength of evidence in the findings and how you then move to relating the findings to a the implications both for research and for practice.
--	--

VERSION 1 – AUTHOR RESPONSE

Reviewer 1: Deirdre Thornlow	
Thanks for the opportunity to review. Well written article on a very important topic.	Thank you very much. We agree that it is an important topic that is under researched.
In the methods section (screening), I would like to have seen higher percentage of two reviews (i.e., 7% with only 73% agreement seems insufficient).	We agree that, although the kappa was acceptable (that is above .6), it would have been ideal to have higher percentages in both reviewing and in agreement. We suggest that the reasons for the lower numbers were because it was a very large number of articles to review and because it had very broad facets: management, quality & safety, and hospitals. We had originally planned for the second reviewer to do at least 10% but due to unforeseen circumstances the second reviewer only had time to complete 7%. When discussing our differences we had the same issue coming up more than once, therefore the discrepancy is not as problematic as it may appear. We have added this point to the text under the screening section: “The primary reoccurring difference in agreement was regarding whether the article pertained to quality of care, owing to the broad nature of the definition.”

	We also replace “A good agreement inter-rater reliability score” with “A moderate agreement inter-rater reliability score”
Nice diagram of screening process (Figure 1).	Thank you.
I found the conclusion to be contradictory as stated, "Managerial activities that affect quality performance are especially highlighted by this review,...However, studies additionally show that many of these contextual factors and activities are lacking..."	We acknowledge that the way we had written this was confusing and have re-written the conclusion to clarify: “Positive actions to consider include the establishment of a Board quality committee.....an infrastructure for staff-manager interactions on quality strategies. However, many of these arrangements were not in place within the study samples.”
Also, several conclusions are based on only 1 or 2 studies.	Thank you, we accept this point. We have changed our wording to avoid over-inflation of one of the conclusions. Instead of “The review suggests”, it now states “More than one study suggest”. We realise the word ‘indications’ is ambiguous; we have removed the following statement from the conclusion (and key messages), so as not to mislead the readers that there were many studies that identified this as an issue: “There are indications of a need to develop managerial knowledge and training on this topic.”
Reviewer 2: Kevin Kenward	
Literature reviews are important due to the ever-increasing output of publications researchers cannot be expected to examine in detail every single new paper relevant to their interests. Thus, it is both advantageous and necessary to rely on regular summaries of the recent literature. The emphasis here is on the word "recent." The authors never give an explanation as to why they reviewed articles only up to the year 2010. That means their most recent article reviewed is four years old by now. Researchers must survey the field and its trends – and this means right up to the moment.	Thank you, we agree with the importance of the need for literature reviews and the need for it to be currently relevant. Please see our response to the editor above. In summary, we understand this point and the simple reason for the time passing between the search date and current date is simply because it was such a large review (over 15,000) and that the entire systematic review process took time to carry out comprehensively. We would argue that, because the review includes literature going back to 1983, and because there are not many empirical papers being written on this topic, that this run date should not impact greatly on the usefulness of this paper. We add to the limitation of the manuscript that: “Due to the enormity of this review, the publication of this article is some time after the search run date. As there is little evidence published on this topic, we consider this not to greatly impact on the current relevance of the review, particularly as the literature reviewed spans almost three decades.”
For literature reviews to be useful they need to be compiled in a professional way. The researchers	Thank you, we spent a great deal of time to ensure that the review was as thorough and scientific as

seem to have been comprehensive in their search having identified and reviewed over 15,000 articles. The authors also attempt to assess the quality of the 19 articles selected for review by using an assessment tool. However, no explanation is given as to the assessment measures or how to interpret them. It would be helpful to know what a "good" score is since the scores range from 55% to 100%.	possible and we are grateful that this is recognised. The tool that we used to assess the quality of the studies by Kmet (2004) does not specify cut-off points for good or poor scores. Yet they do imply this with examples for cut-off points for studies to be considered good enough to include within the review: providing 75% as an example of a conservative threshold and 55% as an example of a liberal threshold. Only one of our included articles scored lower than the liberal cut-off (50%). We decided to include all articles because of the room for error in subjective quality assessment scoring, however we have now added the following statement: “one study scored (what we consider to be) very low (i.e. <55%), eight studies scored highly (i.e. >75%), two other articles scored highly on one out of two of their studies (quantitative/qualitative), and the remaining eight scored a moderate rating in-between.” As way of explanation of the assessment measure, we further provide an example of the rating criteria: “Box 1 shows an example definition of what constitutes ‘Yes’ (2), ‘Partial’ (1) and ‘No’ (0) rating criteria.” Please see box 1 for the full description. We also add more information on the quality of the articles, please see tracked changes for this.
Table 1 presents a good summary of the 19 articles. In addition to or substitution of this table could be a table that indicates which studies address which of the five research questions. For instance, the studies that relate to the question of time managers spend on time and safety could be listed or presented in summary form.	Thank you and we are grateful for this suggestion. We have added an extra column to Table 1 that states whether the article findings refer to our five research questions: time spent, activities, impact, engagement or contextual factors.
The definition of "manager" seems rather broad. The review would also have benefited from a discussion about the different managerial roles and how those roles relate to quality and safety initiatives. For example, Boards set policies and middle managers implement policies. The different roles and responsibilities would then impact how much time was spent on quality activities.	Unfortunately there is no universal definition of a manager and especially of a healthcare manager. We have attempted to tighten up the way we have worded our definition: “A manager was defined as an employee that has subordinates, oversees staff, is responsible for staff recruitment and training, and holds budgetary accountabilities.”

	We have added the following sentences to the discussion section, building on a finding that we raised in the results section: “Only one study clearly demonstrated that senior management and Board priorities can impact upon middle management quality-related activities and engagement. Considering the likely influence that seniors have on their managers, examination of the interactions between the different roles held (e.g. Boards setting policies on quality and middle managers implementing them) would improve our understanding of how these differences reflect in their time spent and actions undertaken.”
There also could have been more critical analysis of the distinction between authors' interpretation of their data and the actual empirical evidence presented. A good review critically analyses how accurately previous authors have reported their findings and whether they have refrained from asserting conclusions not supported by data. Without going through the summary of the 19 articles it is difficult to determine when a statement is being supported by qualitative or quantitative research.	As part of Kmet's quality assessment questions, we checked the following quality criteria for all articles: 'conclusions supported by the results?'. However, because we did not report the articles' conclusions but only their findings, this should not present a problem within our manuscript. We have added the following: “all but three studies..asserted conclusions clearly supported by the data.”
The authors innovative in presenting the theories and results using the input process output (IPO) model. Although the model is presented in one short paragraph and a diagram. There is no discussion as to how to use the model or what insights it provides.	We are glad that the IPO model is considered a useful representation of the research. We add to the discussion on what the IPO presents, please see tracked changes for the long addition. We also add to the conclusion: “We present the IPO model to summarise the evidence-based promotion of conditions and activities in order to guide managers on the approaches taken to influence quality performance.”
Reviewer 3: Elizabeth West	
This is a really interesting paper and has great potential as a springboard for further research. I would like to raise some points for clarification and qualify some of my answers to the checklist above.	We are very pleased to hear this, thank you. We hope that further research will use our review as a helpful starting point.
1. Abstract: could you be more specific about what the actual findings of the review were? I would also like to have a more complete summary of the findings at the end of the paper, immediately before the discussion. The discussion does bring in a lot of new contextual material which perhaps should be cited and discussed at the beginning of the paper and could then be used in the discussion section to relate	We have updated the abstract's results and conclusion sections to add more information on the findings. We have limited examples of findings due to the word count restriction. We intended that the IPO model at the end of the

the findings of the review to this literature.	results section to be the summation of key points from the findings. We add further text to accompany the diagram and present what the IPO model offers. Within the discussion, we have only made claims derived from the evidence within the review or included literature supporting it (please see our response to comment 10). We acknowledge that we did not make this very clear.
2. This is a review of mainly descriptive studies. The authors have conducted a systematic and extensive search but I would like to see more attention to and critique of the main research designs employed in this body of literature.	Thank you for this suggestion. Within the results, we have added more detail on study designs and further information regarding the quality of the studies. We have also added to the critique of the research designs within the discussion. There were many additions for this throughout the article so please refer to tracked changes on the manuscript for these.
3. In the first paragraph of “methods” can you please spell out the definitions of “quality of care” and “patient safety” that you are using so that the reader has them to hand?	Thank you, rather than just referencing the IOM & ARQH citations as we had done previously, we have now added the full definitions to the text.
4. I am not sure that the definition of “manager” on page 6 is entirely consistent with the description of the study participants in table 1, e.g. inclusion of unit nurse managers.	Unit nurse managers fit (depending on the hospital structure) under the definition of frontline managers or middle managers. Where the article has stipulated them as a certain tier of manager (e.g. frontline or middle), we have reported them accordingly. We have also tightened up the wording of the definition for a manager.
5. Why did you exclude mental health acute settings?	We excluded these because we consider mental health hospitals to be specialist centres, where the roles of managers are likely to be quite distinct around the topic of quality, and we wanted to keep the sample as homogenous as possible, considering it is such a broad population sample. We have added this following explanation in parentheses: “(in order to keep the sample more homogenous)”.
6. What is the technical meaning of “facets” at the bottom of page 6?	Some people refer to facets as concepts. They are conceptual groupings in order to make the search strategy easier to design and easier to understand by the reader. It allows for logical grouping of the search terms. We understand people may not be familiar with this term. Following the term “facet” we have now added in brackets “(i.e. a conceptual grouping of related search terms)”.
7. Were any articles not included in the study because they were not of sufficiently high	Thank you for pointing this out. We included all

quality?	articles regardless of their quality scores. The quality assessment is quite subjective and our assessment tool did not specify appropriate cut-off points for inclusion/exclusion. Because of this we chose not to exclude articles on the basis of their scores. To make it clear, we have added the following text under 'methodological quality': "All studies were included regardless of their quality scores." We have also added that scores >55 are considered by us to be very low.
8. What was done about the 27% of articles about which there was disagreement between reviewers (top of page 8)?	We have added the following to answer this question: "Each article was discussed individually until a consensus was reached on whether to include or exclude."
9. It would be good to have more discussion of the typical research designs used in this body of literature as well as their overall quality.	Please see our response to your comment (number 2) above.
10. Summarise the findings and discuss the strength of the evidence. Is it a problem that in table 1, most of the outcomes are "perceptions" rather than patient outcomes? If managers spend a lot of their time on quality issues, would you not expect that they would then report that this has a significant impact on these issues? Is there evidence in this paper for the wider claims emerging from this paper, in, for example, paragraph 2 of the discussion and there on to the end of the paper?	We agree with your point regarding perceptions. We had outlined in our limitations that because the majority of findings are based on self-perceptions, social desirability may have resulted in exaggerated processes and inflated outputs. We have now added here the general concern of the studies focusing on perceptions. "There is an over-reliance on perceptions across the studies, which ultimately reduces the validity of the conclusions drawn from their findings." Earlier in the discussion we also emphasise the point that there is little focus on objective outcomes, by adding the following text: "This evidence is further weakened by the largely descriptive nature of many of the studies. They most lack theoretical underpinnings and appropriate objective measures. Very few studies reported objective clinical quality outcome measures that better show the influence of managerial actions." We add this point to the abstract and key messages to emphasise the importance of it. In the discussion we have intended to make claims and implications that have derived only from the evidence itself and from the literature beyond the included articles from the review. We chose not to re-reference them again because we have already cited them in our results section. Instead we only report citations in the discussion that are from literature that is not included within our review. We have seen that

	this is common practice in other articles. We can see how this might have been unclear. To make it clearer, we have amended the following: “Evidence promotes..” with “Evidence from the review promotes..”.
Because I think that this is a potentially important paper, I have highlighted issues of research design in the review. I think it is important that the reader knows more about the kind of studies that are included; this then informs how you discuss the strength of evidence in the findings and how you then move to relating the findings to a the implications both for research and for practice.	Thank you for your comments; we believe these comments have strengthened the paper.

VERSION 2 – REVIEW

REVIEWER	Kevin Kenward Director of Research Health Research & Educational Trust (HRET) USA
REVIEW RETURNED	08-Jul-2014

- The reviewer completed the checklist but made no further comments.

REVIEWER	Elizabeth West University of Greenwich UK
REVIEW RETURNED	09-Aug-2014

GENERAL COMMENTS	The authors have fully addressed all the points raised by reviewers in the first round of reviews. The review is now more critical and incisive, particularly about the quality of the articles reviewed. This paper makes a valuable contribution to the literature and is very timely, given current concerns about the quality of care. It suggests that there is an urgent need for further investigations of the managerial contribution to quality in the NHS, linking managers actions to patient outcomes. Thank you for addressing the reviewers comments so clearly and comprehensively. I learned a great deal from this manuscript.
--